# Reducing Ship's Energy Consumption through Accommodation and Cargo Spaces Lights Automation

**Dilyan Dimitranov and Blagovest Belev \***

Department of Navigation, Nikola Vaptsarov Naval Academy, 9002 Varna, Bulgaria
* Correspondence: bl.belev@naval-acad.bg

**Abstract:** The effective use of energy proposes using less energy to achieve the same goal. The International Maritime Organization started using criteria for ship's energy efficiency in 2013, when the new International Convention for the Prevention of Pollution from Ships, Annex VI, was adopted. The purpose of the Annex is to improve the ships' energy efficiency through the use of new technologies such as improved hull design, and new propulsion systems, including innovations in energy management. One of the topics for improving the car carriers' energy efficiency is the constantly working lights in the superstructure and on the car decks. Lights' energy consumption may be negligible on most ships, but on car carriers, the consumption is of greater magnitude. This article presents a survey of the improvement of energy consumption on board a car carrier during regular voyages as well as the cost-effectiveness of introducing light control automation. The authors of this article review an example of light automation set up to control the lights in a ship's superstructure and on the car decks. The implementation of such a system and the different types of automation are also reviewed. This research is towards new regulations, established by Maritime Environment Protection Committee and implemented in shipping since 1 November 2022. Conclusions for practical use are extracted.

**Keywords:** ship's energy efficiency; energy consumption control; LED lights automation; fuel savings; motion detection

## 1. Introduction

Shipping around the world produces an enormous amount of harmful gases such as nitrous oxides, sulfur oxides, particulate matter, and carbon dioxide. Shipping has been responsible for around 20% of the total worldwide $CO_2$ emissions [1]. That is the reason why ships are the biggest air polluters in ports.

Most containers and cruise ships as well as tankers and cargo vessels burn large amounts of heavy fuel oil. Together 90,000 ships consume 370 million tons of heavy fuel oil and emit 20 million tons of nitrous oxides into the atmosphere [2]. However, when a vessel navigates the inner waterways it uses light diesel oil, which is significantly less harmful to the environment.

The problem is expected to develop further if no action is performed. "Maritime forecast to 2050", created by DNV GL experts says, that air pollution is expected to increase by at least 50% in the best-case scenario and up to 250% in the worst-case scenario by 2050 [2]. The conclusion from a study by the European Parliament and The International Convention for the Prevention of Pollution from Ships (MARPOL), Annex VI, is that shipping will be responsible for a fifth of the $CO_2$ global emissions [3].

Energy efficiency has become a topic for a lot of discussions recently, by merit of its impact on the economy and on the marine environment. For several years now, ships began a process to significantly increase their energy efficiency through the use of speed optimization, weather routing, hull monitoring and maintenance, and smart management of the electrical grid. Most of the newly built vessels use light-emitting diode (LED)

technologies to provide artificial light, because of the significant benefits of LED over fluorescent and halogen light bulbs. Even though LED lights are far less energy-demanding, the use of light automation systems can further increase their energy efficiency.

There are not many studies concerning the vessels' lighting systems. Uddin examines the power quality voltage sags' effects on LEDs [4]. Aman developed a study on the performance of domestic lighting based on incandescent lamps, fluorescent lights (FL), and LEDs [5]. Gan and Fuchtenhans have studied the problem with the electrical and photometric attributes of LEDs and FLs [6,7]. Only a few of the researchers have turned their focus toward the benefits of using LED lights on board ships [8]. Su et al. have proposed a performance evaluation model for cost-effective light equipment on ships [9]. Mills, Gengnagel, and Wollburg have looked into the technical and economic gains when fishermen replace their kerosene lanterns with LED light during night fishing and Yigit, Kökkülünk, and Savas examine the economic values and the environmental impact of all the ship's lights are replaced with LED technology [10,11].

Smart lighting is a comparatively new research topic, that emerged after serious technological development. After all, about 19% of the energy consumption around the world is attributed to lights [12]. Fossil fuel usage for energy conversion has significantly increased in the last decades and with that, the share of lighting in global carbon dioxide emissions has also increased [13]. The share of lighting costs depends on the company type and the effectiveness of the light technology the company is currently using [14]. Advancements in LED technology were the precursor to smart lighting systems. The new technology enables fixtures to be controlled with relative ease to shed light only when and where it is needed. Smart lights can be easily combined with various types of external sensors to respond to a fast-changing environment. Linking individual lights via a bridge network allows for the inclusion of external sensors and for the incorporation of specific algorithms to further improve their effectiveness [15]. When added together, all these aspects offer the provision of extra value, such as light communication, internal positioning, etc. [14]. On an industrial scale, smart lighting has the capacity to raise energy efficiency with up to 25% [6]. In that regard, Steinbacher proposed an algorithm for light automation based on the movement of the service cars on a container terminal with the goal to decrease light energy consumption [16]. Furthermore, energy monitoring can lead to improvements to the maintenance schedule of light fixtures [17].

Since 01 of November 2022, new regulations regarding Ships' Energy Efficiency Management Plan (SEEMP) are in force. Maritime Environment Protection Committee (MEPC) at its 78th Session adopted mandatory goal-based technical and operational measures to reduce the ship's carbon intensity. A new part III in the SEEMP is created and results of statistics on yearly basis have to reflect in order to reduce carbon emissions [18,19]. Classification society members do not have regulations regarding deck lights and all other lights in the ship's public area and they advise any innovation with respect to the above to be a part of SEEMP, Part III [20–22].

A survey on the improvement of energy consumption on board a car carrier is made in this article and the cost-effectiveness of light control automation is introduced, as well. The authors review an example of light automation set up to control the lights in a ship's superstructure and on the car decks. A possible implementation of such a system and the different types of automation are also reviewed. All artificial light sources have been cataloged and their individual and total energy consumption has been established on a daily basis, as well as their expected energy savings through the use of the proposed setup. The aim of the article is to propose a practical approach to efficient control of the ship's lights. A compared analysis for oil consumption of diesel generators is performed—in the case of automation method implementation and in the conventional method.

## 2. Materials and Methods

Two types of surveys were made on board a real car carrier ship between destinations in Japan and the Middle East. The first survey focuses on the necessary time for the lights

to be turned on in the superstructure and on the car decks. The second survey is about the energy consumption of the same lights. The research took place between 1 June 2021 and 30 August 2021 when the average daylight time was about fifteen hours.

Data for calculations were collected from all lights in all ship's compartments. Most of the information concerns car decks, which are not lit from outside and need artificial light. Every compartment is described from the view of its working dedication. The focus is on safety and electricity consumption is not with priority against crew and ship safety. All the calculations are made on daily, weekly, and monthly basis.

## 2.1. LED Lights Energy Consumption in the Superstructure

All rooms and compartments in the superstructure are listed below as well as observations about how often artificial light is required daily. All compartments, listed below, are located in Figure 1.

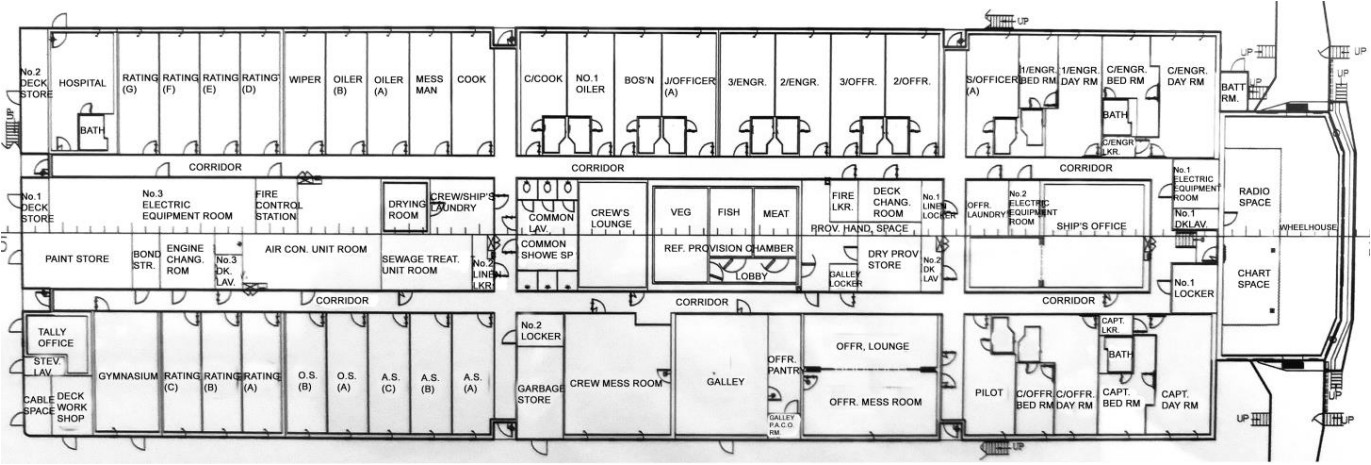

**Figure 1.** Schematic of a car carrier superstructure.

1. Wheelhouse—the wheelhouse is used all the time for watch keeping and there is always an officer on the bridge. The artificial lights are used rarely, and it is hard to develop a light optimization model. Therefore, this compartment will not be reviewed in terms of energy consumption savings.
2. Ship's Office—there are 12 LED lights, 40 W each, divided into 6 pairs inside. The office is used every day during navigation, usually between 07:30 and 17:00. In the ship's office of this vessel there is no access to natural light, so artificial light is used whenever there are people inside. The office is primarily used by the chief officer for toolbox meetings, cargo stowage planning, and ballast operations. Seagull training computers, which are also located in the office are used outside of working hours by all members of the crew. When the vessel is in port, the office is used all the time for meetings with port authorities, agents, and stevedores, trim and heel monitoring, ventilation, and ballast operations. During normal navigation, usually, it is not necessary for the lights to be on more than 8 h a day.
3. Stationary—the room is used to store all stationary articles and is lit by one 20 W LED light. There is no access to natural light. The stationary is entered only for the pickup of items and for planned inventory. The light is not used more than 3 h a week.
4. Linen Locker 1 and 2—the lockers are lit by one 20 W light each and there is no access to natural light. The lockers house bedsheets, blankets, pillows, and other linen products. The lights are not used more than 3 h a week.
5. No1 Electric Equipment Room—the locker is lit by one 20 W LED light and there is no access to natural light. The locker contains spare parts for equipment in the superstructure. The light is not used more than 3 h a week.

6.　No3 Electric Equipment Room—the room is lit by eight 40 W LED lights, split into 4 pairs and there is no access to natural light inside. The breakers for the lights in the accommodation and the lights on the car decks are located here, as well as the manual control of the ship's ventilators. The room is accessed for weekly checks and for vent operation when the ship is in port. The lights are not used more than 6 h a week during voyages.

7.　Paint Store—the store is lit by four 40 W LED lights divided into 2 pairs. There is access to natural light through the door, but the light is not enough in the far corner of the store, so artificial light is used as well. The store is also used for paint preparation and mixture before every paint job, apart from the paint storage. The lights are not used more than 1 h a day.

8.　No1 Deck Store—the store is lit by one 20 W LED light and there is no access to natural light. The store houses spare parts and spare instruments for the ship's daily maintenance. The light is not used more than 1 h a day.

9.　Emergency Diesel Generator Room—the room is lit by four 40 W LED lights, divided into 2 pairs. There is access to natural light through the door, but it is insufficient in the far corners of the room, so artificial light is used as well. The room houses the emergency diesel generator and is used for weekly checks and monthly drills. The lights are not used more than 2 h a week.

10.　Bond Store—the store is lit by one 20 W LED light and there is no access to natural light. The store contains liquor and cigarettes. The light is not used more than 4 h a week.

11.　Deck Workshop—the workshop is lit by two 40 W and there is access to natural light through the door, but the light is insufficient in the far corners of the room. The workshop house all the tools and equipment required for the daily maintenance of the vessel, and it used for small repairs. The lights are not used more than 3 h a day.

12.　Gymnasium—the gym is lit by eight 40 W LED lights, divided into 4 pairs and there is access to natural light through a window. However, the natural light is insufficient for the safe practice of various exercises and artificial light is used in conjunction with natural light. The gym is used by at least half the crew and the lights are not used more than 6 h a day.

13.　Hospital—the hospital is lit by eight 40 W LED lights, divided into 4 pairs and there is access to natural light through a window. However, the natural light is insufficient for any kind of medical manipulations. The lights in the hospital are not used more than 2 h a week.

14.　Laundry Rooms 1 and 2 (Crew and Officers)—the rooms are lit by two 40 W LED lights each and there is no access to natural light. The washing machines inside are used every day, but the lights are not used more than 3 h a day.

15.　Drying Room—the room is lit by one 20 W LED light and there is no access to natural light. The room is heated and used to quickly dry washed clothes. The light is not used more than 3 h a day.

16.　Recreation and Mess Rooms 1 and 2 (Crew and Officer)—the rooms are used for dining and rest by the crew. They are lit by twelve 40 W LED lights each, grouped in 6 pairs. There is access to natural light through the windows which is enough for all purposes in the rooms. The lights are not used for more than 6 h a day.

17.　Galley—the room is lit by eight 40 W LED lights divided into 4 pairs and there is access to natural light through a window, but the natural light is not enough for safe cooking. Due to the nature of the room, the lights are used for about 16 h a day.

18.　Common Showers and Common Lavatories—the rooms are lit by four 40 W LED lights and there is no access to natural light. The lights are not used for more than 3 h a day.

19.　Fire Station—the room is lit by two 40 W LED lights and there is no access to natural light. The room is used for control of the firefighting systems on board and as a store for firefighting equipment. The lights are not used more than 2 h a week.

20.  Refrigeration Provision Chambers—the chambers consist of a hallway, fish chamber, meat chamber, and vegetable chamber. The fish and meat chambers are lit by a single 20 W LED light each. The hallway and the vegetable chamber are lit by two 40 W LED lights each. The lights are not used more than 2 h a day.
21.  Changing Rooms 1 and 2 (Deck and Engine)—the rooms are lit by two 40 W LED lights and there is no access to natural light. The lights are not used for more than 3 h a day.
22.  Tally Office—the room is lit by two 40 W LED lights and there is access to natural light through a window. The room is used mainly by stevedores when the vessel is in port and rarely used during voyages. The lights are not used more than 1 h a day.
23.  Garbage Store—the store is lit by two 40 W LED lights and there is access to natural light through the door, but it is insufficient for the far corners of the room. The lights are not used for more than 1 h a day.
24.  No2 Deck Store—the store is lit by one 20 W LED light and there is access to natural light through the door. The store contains spare ropes and wires. The light is not used for more than 2 h a week.

In Table 1 are ordered the compartment and their light consumption per day.

**Table 1.** Rooms and compartments subject to the research.

| Compartment | Number of Lights | Power [W] | Working Hours/Day |
| --- | --- | --- | --- |
| 1 | - | - | - |
| 2 | 12 | 40 | 8 |
| 3 | 1 | 20 | 0.4 |
| 4 | 1 | 20 | 0.4 |
| 5 | 1 | 20 | 0.4 |
| 6 | 8 | 40 | 1 |
| 7 | 4 | 40 | 1 |
| 8 | 1 | 20 | 1 |
| 9 | 4 | 40 | 0.3 |
| 10 | 1 | 20 | 0.6 |
| 11 | 2 | 40 | 3 |
| 12 | 8 | 40 | 6 |
| 13 | 8 | 40 | 0.3 |
| 14 | 4 | 40 | 3 |
| 15 | 1 | 20 | 3 |
| 16 | 36 | 40 | 6 |
| 17 | 8 | 40 | 16 |
| 18 | 4 | 40 | 3 |
| 19 | 2 | 40 | 0.3 |
| 20 | 2 + 4 | 20 + 40 | 2 |
| 21 | 2 | 40 | 3 |
| 22 | 2 | 40 | 1 |
| 23 | 2 | 40 | 1 |
| 24 | 1 | 20 | 0.3 |

When the lights in the listed rooms and stores work all day and night, they consume 4360 W of electricity every hour. Hence:

$$4360 \text{ W} \times 24 \text{ h} \div 1000 = 104.64 \text{ kWh per 24 h} \tag{1}$$

The energy consumption of the lights is 104.64 kWh per day.

*2.2. Car Decks Lights*

The Safety Management System of the vessel's operator has not implemented a procedure to describe if and/or when the lighting systems on board should be switched off. Therefore, all lights on the car decks are permanently switched on.

The lights on the car decks can be divided into two categories:

- Stairway lighting fixtures, which are equipped with 20 W LED lights divided into pairs.
- Car deck lighting fixtures, which are equipped with 40 W LED lights divided into pairs.

The thirteen-car decks are connected to each other through rampways and staircases. The rampway lights are included in the total number of lights for each deck.

There are four independent staircases on board, which provide access to the car decks. The staircases are lit by 141 LED lights. When the lights work continuously, they consume 2820 W of electricity every hour. Hence:

$$141 \times 20 \text{ W} = 2820 \text{ W} \tag{2}$$

$$2820 \text{ W} \times 24 \text{ h} \div 1000 = 67.68 \text{ kWh per 24 h} \tag{3}$$

The energy consumption of the lights is 67.68 kWh per day.

The lights on the car decks are distributed as follows (Table 2):

**Table 2.** Distribution of lights on the car decks.

| Decks | Lights in Hold 4 | Lights in Hold 3 | Lights in Hold 2 | Lights in Hold 1 | Total |
|---|---|---|---|---|---|
| Garage | 6 rows of 12 | 6 rows of 12 | | | 144 lights |
| Deck 12 | 7 rows of 12 | 7 rows of 12 | 7 rows of 12 | 7 rows of 12 | 336 lights |
| Deck 11 | 7 rows of 12 | 7 rows of 12 | 7 rows of 12 | 7 rows of 12 | 336 lights |
| Deck 10 | 7 rows of 12 | 7 rows of 12 | 7 rows of 12 | 7 rows of 12 | 336 lights |
| Deck 9 | 7 rows of 12 | 7 rows of 12 | 7 rows of 12 | 7 rows of 12 | 336 lights |
| Deck 8 | 7 rows of 12 | 7 rows of 12 | 7 rows of 12 | 7 rows of 12 | 336 lights |
| Deck 7 | 7 rows of 12 | 7 rows of 12 | 7 rows of 12 | 7 rows of 12 | 336 lights |
| Deck 6 | 6 rows of 12 | 7 rows of 12 | 7 rows of 12 | 7 rows of 12 | 324 lights |
| Deck 5 | 7 rows of 12 | 7 rows of 12 | 7 rows of 12 | 6 rows of 12 | 324 lights |
| Deck 4 | | 7 rows of 12 | 7 rows of 12 | 4 rows of 12 | 216 lights |
| Deck 3 | | 6 rows of 12 | 6 rows of 12 | 3 rows of 12 | 180 lights |
| Deck 2 | | 6 rows of 12 | 6 rows of 12 | 2 rows of 12 | 168 lights |
| Deck 1 | | 5 rows of 12 | 5 rows of 12 | | 120 lights |
| | | | | Grand total: | 3492 lights |

Each light is of 40 W. Therefore, when the lights work continuously, they will consume:

$$3492 \text{ lights} \times 40 \text{ W} = 139{,}680 \text{ W} \tag{4}$$

$$139{,}680 \text{ W} \times 24 \text{ h} \div 1000 = 3352.32 \text{ kWh per 24 h} \tag{5}$$

*2.3. Analysis of the Electric Power Generation and Its' Fuel Consumption*

During the months of June, July, and August 2021 the following data have been obtained for the ship's diesel generators (Table 3):

**Table 3.** Diesel generator data table for the months of June, July, and August 2021.

| JUNE | | | | JULY | | | | AUGUST | | | |
|---|---|---|---|---|---|---|---|---|---|---|---|
| **Data** | **1** | **2** | **3** | **Data** | **1** | **2** | **3** | **Data** | **1** | **2** | **3** |
| 01.06 | 437 | 2.6 | 4.034 | 01.07 | 433 | 2.6 | 3.997 | 01.08 | The vessel is in port | | |
| 02.06 | 430 | 2.5 | 4.128 | 02.07 | 421 | 2.6 | 3.886 | 02.08 | The vessel is in port | | |
| 03.06 | 228 | 2.6 | 2.105 | 03.07 | The vessel is in port | | | 03.08 | The vessel is in port | | |
| 04.06 | 443 | 2.5 | 4.253 | 04.07 | The vessel is in port | | | 04.08 | 707 | 2.9 | 5.851 |
| 05.06 | 535 | 2.6 | 4.938 | 05.07 | 586 | 4.5 | 3.125 | 05.08 | 654 | 2.9 | 5.412 |
| 06.06 | 450 | 2.5 | 4.320 | 06.07 | 590 | 2.6 | 5.446 | 06.08 | 632 | 2.9 | 5.230 |
| 07.06 | 441 | 2.6 | 4.071 | 07.07 | The vessel is in port | | | 07.08 | 610 | 2.8 | 5.229 |
| 08.06 | 470 | 2.6 | 4.338 | 08.07 | The vessel is in port | | | 08.08 | 600 | 3 | 4.800 |
| 09.06 | 491 | 2.8 | 4.209 | 09.07 | The vessel is in port | | | 09.08 | 607 | 2.8 | 5.203 |
| 10.06 | 505 | 2.7 | 4.489 | 10.07 | The vessel is in port | | | 10.08 | 615 | 2.9 | 5.090 |
| 11.06 | 523 | 2.8 | 4.483 | 11.07 | The vessel is in port | | | 11.08 | 604 | 2.8 | 5.177 |
| 12.06 | 510 | 2.5 | 4.896 | 12.07 | 600 | 2.6 | 5.538 | 12.08 | 590 | 3 | 4.720 |
| 13.06 | 518 | 2.8 | 4.440 | 13.07 | 610 | 2.8 | 5.229 | 13.08 | 611 | 2.9 | 5.057 |
| 14.06 | The vessel is in port | | | 14.07 | The vessel is in port | | | 14.08 | The vessel is in port | | |
| 15.06 | 520 | 2.6 | 4.800 | 15.07 | The vessel is in port | | | 15.08 | The vessel is in port | | |
| 16.06 | 720 | 2.9 | 5.959 | 16.07 | The vessel is in port | | | 16.08 | 593 | 2.9 | 4.908 |
| 17.06 | 463 | 2.6 | 4.274 | 17.07 | 600 | 3 | 4.800 | 17.08 | 614 | 2.8 | 5.263 |
| 18.06 | 412 | 2.7 | 3.662 | 18.07 | 620 | 2.9 | 5.131 | 18.08 | 601 | 3 | 4.808 |
| 19.06 | 421 | 2.5 | 4.042 | 19.07 | 640 | 2.9 | 5.297 | 19.08 | 615 | 2.9 | 5.090 |
| 20.06 | 404 | 2.6 | 3.729 | 20.07 | 640 | 2.9 | 5.927 | 20.08 | 608 | 2.9 | 5.032 |
| 21.06 | The vessel is in port | | | 21.07 | 615 | 3 | 4.920 | 21.08 | 621 | 3 | 4.968 |
| 22.06 | The vessel is in port | | | 22.07 | 645 | 2.8 | 5.529 | 22.08 | The vessel is in port | | |
| 23.06 | The vessel is in port | | | 23.07 | 650 | 2.9 | 5.379 | 23.08 | The vessel is in port | | |
| 24.06 | The vessel is in port | | | 24.07 | The vessel is in port | | | 24.08 | The vessel is in port | | |
| 25.06 | The vessel is in port | | | 25.07 | The vessel is in port | | | 25.08 | The vessel is in port | | |
| 26.06 | The vessel is in port | | | 26.07 | The vessel is in port | | | 26.08 | The vessel is in port | | |
| 27.06 | The vessel is in port | | | 27.07 | The vessel is in port | | | 27.08 | The vessel is in port | | |
| 28.06 | The vessel is in port | | | 28.07 | The vessel is in port | | | 28.08 | The vessel is in port | | |
| 29.06 | The vessel is in port | | | 29.07 | The vessel is in port | | | 29.08 | The vessel is in port | | |
| 30.06 | The vessel is in port | | | 30.07 | The vessel is in port | | | 30.08 | The vessel is in port | | |
| | | | | 31.07 | The vessel is in port | | | 31.08 | The vessel is in port | | |

In the table's header 1 is DG Average Power per 24 h [kW], 2 is diesel oil consumption per 24 h [mt], 3 is consumed electricity per 24 h per 1 ton of diesel oil [MWh].

Data are not provided for the periods when the vessel is in port, due to the increased load of the diesel generators and the mandatory use of all car deck lights.

The average power generated by a diesel generator on a 24-h basis for the three months was 539.8571 kW. The average electricity produced for 24 h was 12,956.57 kWh with average diesel oil consumption of 2.744898 mt per 24 h.

### 2.4. Proposed Method to Reduce LED Light Energy Consumption

In order to reduce the energy consumption of the LED lights in the superstructure and on the car decks, the authors recommend a system of automatic relay switches and motion sensors. Various manufacturers around the world provide such relays and sensors, which can be monitored and controlled via phone applications or computer software. The relays and the sensors can also be automated with additional software. For this recommendation: the relays were programmed to switch on certain lights when certain sensors detect motion.

The relays are controlled via the Wi-Fi signal from the routers in the superstructure. They do not require access to Internet except for the initial installation or for firmware updates.

It is not required to make any structural modifications on the vessel to install the relays, as they can be positioned behind the switch for on/off power in the superstructure. The relays, which control the car deck lights can be installed behind the breakers in the electrical distribution room. The cable connection must follow this diagram (Figure 2):

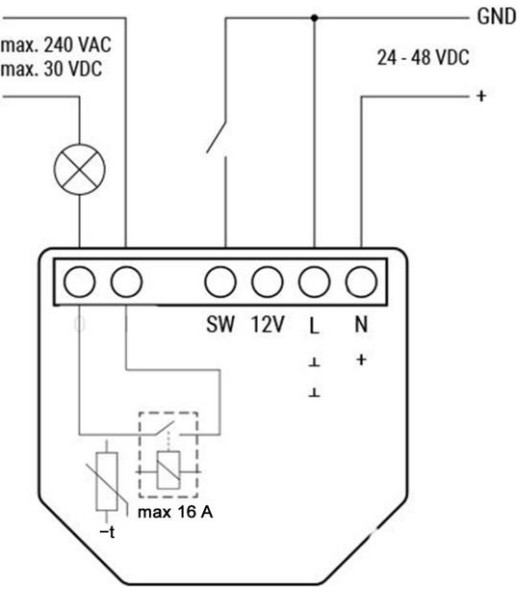

**Figure 2.** Relay wiring diagram.

The proposed relays can be connected to AC or to DC, depending on the electrical network on board the vessel. The relays can also be powered by DC and operate AC electronics, if necessary.

The motion sensors should be passive infrared sensors that measure infrared light radiating from objects in their field of view. This type of sensor is selected as opposed to the active ultrasonic sound wave sensors because, during the rolling of the vessels, various inanimate objects can trigger unwanted lights to turn on, if the latter is used. The motion sensors should be mounted on the car decks access points such as:

- Ramps;
- Stairways;
- doors from other compartments.

The motion sensors must be mounted in such a way as to detect any person who is passing through the access points. Electricity for the motion sensors can be provided from the nearest light fixture. However, a cable network leading to a central control hub must be developed, so that the signal from each sensor is processed and analyzed.

The following algorithm is to be implemented for the lights in the superstructure (Figure 3):

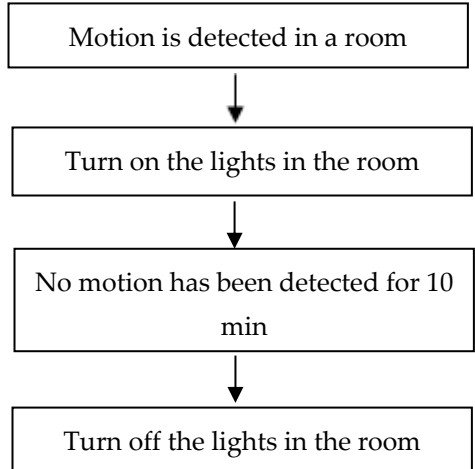

**Figure 3.** Algorithm for lighting the superstructure.

If motion is detected in a room → Turn on the lights in that room.

If the lights are on and no motion has been detected for 10 min → Turn off the lights in the room.

Two types of algorithms are to be implemented for the car decks and stairways (Figures 4 and 5).

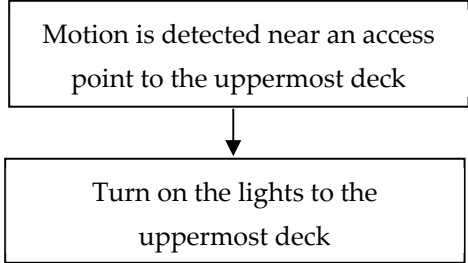

**Figure 4.** Algorithm for car deck access lighting.

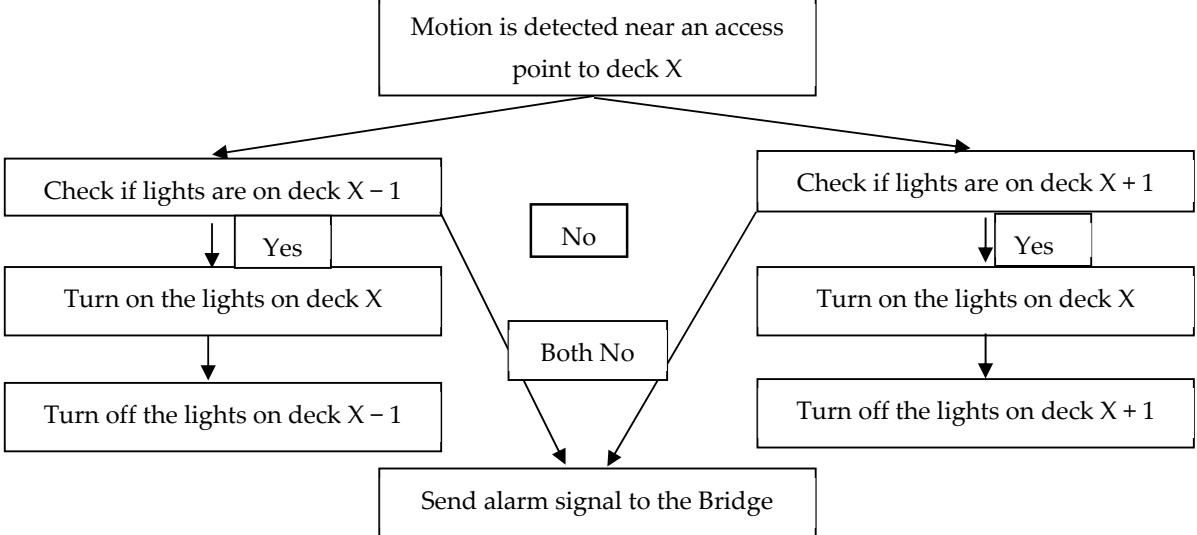

**Figure 5.** Algorithm for car deck lighting.

For the car decks (Figure 4):

If motion is detected near an access point to the uppermost deck → Turn on the lights on the uppermost deck.

If motion is detected near an access point to deck X → Check if the lights are on deck X + 1 or on deck X − 1.

- If either is "Yes", turn on the lights on deck X and turn off the light on deck X + 1 and deck X − 1.
- If both are "No", send alarm to the bridge for unauthorized movement on deck X.

Algorithm for lighting the stairways is shown on Figure 6 and written down.

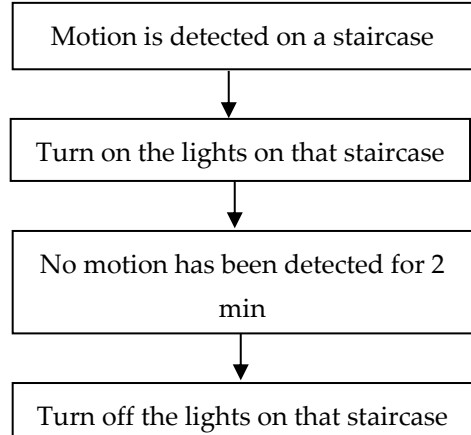

**Figure 6.** Algorithm for staircase lighting.

If motion is detected near a staircase → Turn on the lights on that staircase.

If staircase lights are on and no motion has been detected for 2 min → Turn off the staircase lights.

With the first algorithm, the lights on the triggered car deck remained on, until the person has passed an access point leading to another deck. With the second algorithm the stairway lights were switched on if motion is detected and turned off automatically.

When the vessel is in port all lights are used constantly, hence all automation scripts should be manually switched off through the software or from the central control hub.

In addition, mounting motion sensors on each access point provides an additional level of security against stowaways. Their movement between the car decks after vessel departure will immediately trigger an alarm on the bridge, alerting the officer of the watch.

The introduction of this system shall not replace the existing framework of light control switches. At any moment, any person entering a room or compartment shall be able to manually switch on or off any lights. With this in mind, in case of an automation system failure, the personnel shall be able to use all lighting fixtures and switches as if the automation system was not introduced.

## 3. Results and Discussion

### 3.1. Achievable Reduction in LED Light Consumption in the Superstructure

Using the analysis made in Section 2.1, it can be demonstrated that it is excessive for all of the lights in the superstructure to be switched on all the time. A significant reduction in light usage time can be achieved through the proposed method of light automation in the rooms and stores. The results can be clearly observed in Figure 7, in which the blue bars mark the total hours of practical use of light in the compartment and the orange bars mark the unnecessary hours of light use on a weekly basis.

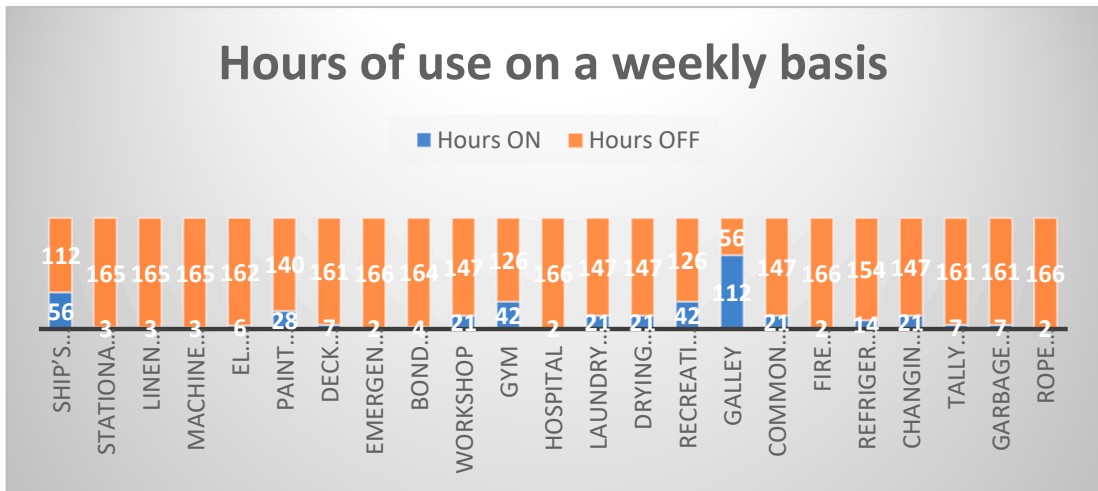

**Figure 7.** Hours of use on a weekly basis.

The substantial reduction in the usage of lights in the superstructure will in turn contribute to less energy usage overall. A comparison between the energy usage in the more commonly visited rooms and stores in the superstructure with and without the proposed automation can be seen in Table 4.

**Table 4.** Use of electricity in everyday rooms and stores with and without automation algorithms in kWh per day.

|  | Ship's Office | Paint Store | Deck Store | Workshop | Gym | Laundry R. 1 and 2 | Drying Room |
|---|---|---|---|---|---|---|---|
| With automation | 3.84 | 0.64 | 0.02 | 0.24 | 1.92 | 0.48 | 0.06 |
| Without automation | 11.52 | 3.84 | 0.48 | 1.92 | 7.68 | 3.84 | 0.48 |
|  | Recreation 1 and 2 | Galley | Crew Showers and Lavatory | Refrigerating Chambers | Changing Rooms | Tally Office | Garbage Store |
| With automation | 5.76 | 5.12 | 0.96 | 0.4 | 0.48 | 0.08 | 0.08 |
| Without automation | 23.04 | 7.68 | 7.68 | 4.8 | 3.84 | 1.92 | 1.92 |

The total amount of used electricity for light fixtures in the more commonly visited rooms and stores in the superstructure for a 24-h period without the proposed automation is the sum of the energy usage in the compartments in Table 4 (6):

$$11.52 + 3.84 + 0.48 + 1.92 + 7.68 + 3.84 + 0.48 + 23.04 + 7.68 + 7.68 + 4.8 + 3.84 + 1.92 + 1.92 = 80.64 \text{ kWh} \quad (6)$$

The total amount of used electricity for the same light fixtures, used for a 24-h period with the proposed automation, is the sum of the energy usage in the compartments in Table 4 (7):

$$3.84 + 0.64 + 0.02 + 0.24 + 1.92 + 0.48 + 0.06 + 5.76 + 5.12 + 0.96 + 0.4 + 0.48 + 0.08 + 0.08 = 20.08 \text{ kWh} \quad (7)$$

This is a 75% reduction in energy use. The disambiguation of this reduction can be viewed in Figure 8.

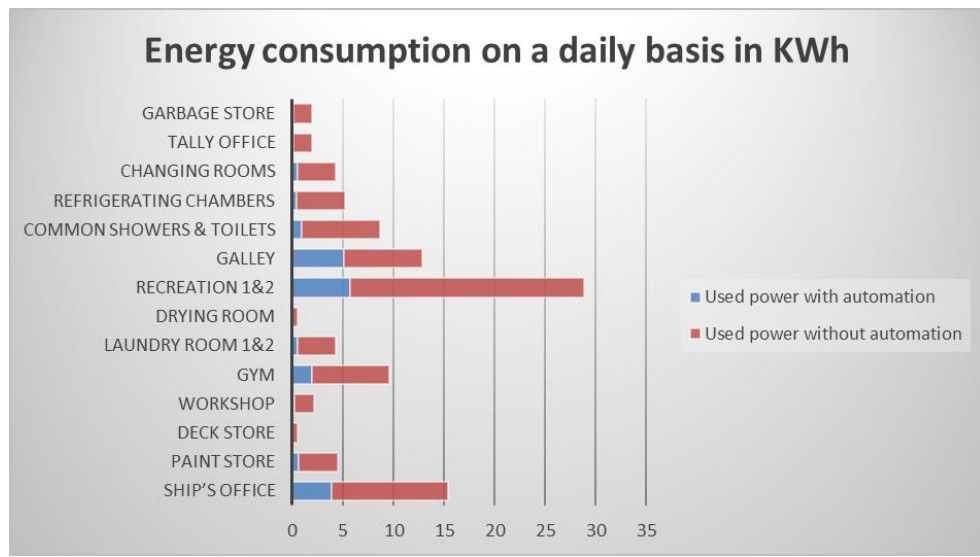

**Figure 8.** Daily energy consumption in KWh.

A comparison between the energy usage in the rest of the rooms and stores in the superstructure with and without the proposed automation can be observed in Table 5.

**Table 5.** Use of electricity in the rest of the rooms and stores with and without automation algorithms in kWh per day.

| | Stationary | Linen Locker 1 and 2 | No 1 Elec. Eq. Room | No 3 Elec. Eq. Room | EDG Room | Bond Store | Hospital | Fire Station | No 2 Deck Store |
|---|---|---|---|---|---|---|---|---|---|
| Used energy with automation | 0.06 | 0.12 | 0.06 | 1.92 | 0.32 | 0.08 | 0.64 | 0.16 | 0.04 |
| Used energy without automation | 2.56 | 5.12 | 2.56 | 40.96 | 20.48 | 2.56 | 40.96 | 10.24 | 2.56 |

The total amount of used electricity for light fixtures for these premises without automation is the sum of the energy usage in the compartments in Table 5 (8):

$$2.56 + 5.12 + 2.56 + 40.96 + 20.48 + 2.56 + 40.96 + 10.24 + 2.56 = 128 \text{ kWh} \tag{8}$$

Consumption of 128 kWh per week equals to 18.29 kWh per day. The total amount of used electricity for light fixtures for these premises with automation is the sum of the energy usage in the compartments in Table 5 (9):

$$0.06 + 0.12 + 0.06 + 1.92 + 0.32 + 0.08 + 0.64 + 0.16 + 0.04 = 3.4 \text{ kWh} \tag{9}$$

Consumption of 3.4 kWh per week equals to 0.49 kWh per day, which is a 97% reduction in energy use. The disambiguation of this reduction can be viewed in Figure 9.

With the proposed method for LED light automation the electricity required to operate the lights can be reduced to:

$$20.08 \text{ kWh} + 0.49 \text{ kWh} = 20.57 \text{ kWh per 24 h} \tag{10}$$

Having in mind calculations (1) the saved amount of electricity is:

$$104.64 \text{ kWh} - 20.57 \text{ kWh} = 84.07 \text{ kWh} \tag{11}$$

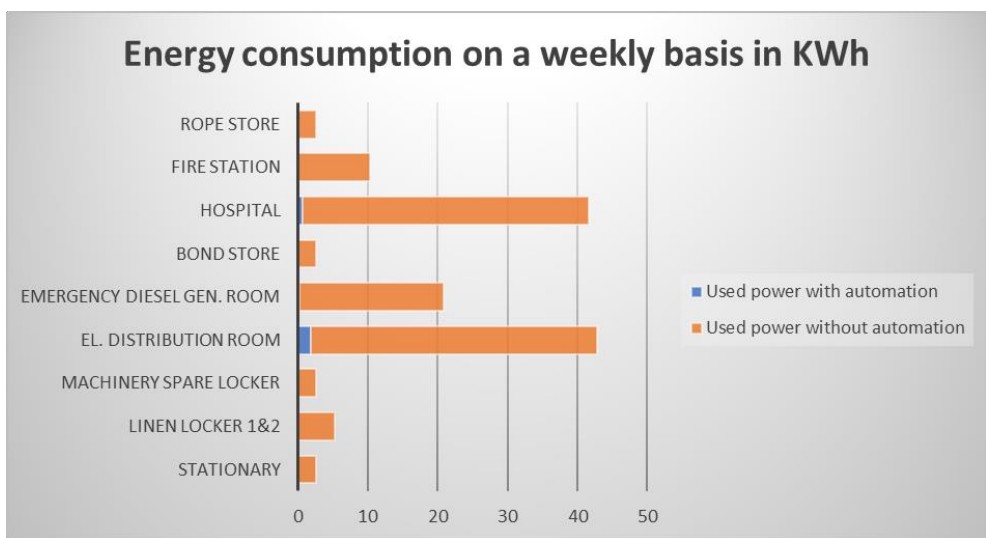

**Figure 9.** Weekly energy consumption in KWh.

*3.2. Achievable Reduction in LED Light Consumption in the Car Decks*

During navigation the stairway lights are use used only when fire patrols are performed. Six fire patrols are performed daily, one in every four hours. Passing through each of the stairways takes less than 30 s. During a single fire patrol, the patrolman will pass a minimum of 13 staircases.

$$13 \times 30 \text{ s} = 390 \text{ s} \tag{12}$$

$$390 \text{ s} \div 60 = 6.5 \text{ min} \tag{13}$$

Therefore, during each fire patrol the stairway lights will be on for at least 6.5 min. For a full day, the stairway lights will be on for 39 min:

$$6.5 \text{ min} \times 6 = 39 \text{ min} \tag{14}$$

This time will be rounded up to 1 h to account for unforeseen circumstances. In this case, the stairway lights will work for 1 h per 24-h period and will consume 2.82 kWh for that period.

During navigation, the lights on the car decks are used only during a fire patrol as well. The time required for a patrolman to pass through a deck of cars is usually around 5 min. The average number of LED lights per deck is 269. Therefore, during a single fire patrol on average 269 LED lights will be continuously switched on.

$$13 \text{ decks} \times 5 \text{ min} = 65 \text{ min} \tag{15}$$

In such a case, 269 LED lights will be working for 65 min during each patrol. As there are 6 patrols per day:

$$65 \text{ min} \times 6 = 390 \text{ min} \tag{16}$$

$$390 \text{ min} \div 60 = 6.5 \text{ h} \tag{17}$$

The 269 car deck lights will be on for 6.5 h in every 24-h period. Their electricity consumption will be:

$$269 \text{ lights} \times 40 \text{ W} = 10,760 \text{ W} \tag{18}$$

$$10,760 \text{ W} \times 6.5 \text{ h} \div 1000 = 69.94 \text{ kWh per day} \tag{19}$$

The achieved reductions will be as follows:

- For the car deck lights:

$$3352.32 \text{ kWh} - 69.94 \text{ kWh} = 3282.38 \text{ kWh} \tag{20}$$

- For the stairway lights:

$$67.68 \text{ kWh} - 2.82 \text{ kWh} = 64.86 \text{ kWh} \tag{21}$$

Total:
$$3282.38 \text{ kWh} + 64.86 \text{ kWh} = 3347.24 \text{ kWh} \tag{22}$$

These accounts show that 3347.24 kWh can be reduced in every 24-h period from LED lights alone. Together with the reductions from the superstructure lights the total amount is:
$$84.07 \text{ kWh} + 3347.24 \text{ kWh} = 3431.31 \text{ kWh per day} \tag{23}$$

*3.3. Achievable Reduction in LED Light Consumption in the Car Decks*

With the proposed automation the average amount of electricity produced by the vessel per day will be:

$$12,956.57 \text{ kWh} - 3431.31 \text{ kWh} = 9525.26 \text{ kWh} \tag{24}$$

To produce 12,956.57 kWh on average per day, 2.744898 tons of diesel oil were used. To produce 9525.26 kWh per day, the amount of diesel oil is 2.018 tons:

$$\frac{9525.26 \text{ kWh} \times 2.744898 \text{ t}}{12,956.57 \text{ kWh}} = 2.018 \text{ t} \tag{25}$$

The saved amount of diesel oil per day is 0.727 tons:

$$2.7449 \text{ t} - 2.018 \text{ t} = 0.7269 \text{ t} \tag{26}$$

If it is assumed that the vessel navigates around 190 days every year, the total diesel oil savings will amount to 138.111 tons.

$$0.7269 \text{ t} \times 190 \text{ days} = 138.111 \text{ t} \tag{27}$$

With an average diesel oil price of USD 1100 per ton, the amount of savings will surmount to USD 151,922.1 every year.

$$138.111 \text{ t} \times 1100 \text{ \$} = 151,922.1 \text{ \$} \tag{28}$$

**4. Conclusions**

The goals of the article are achieved. The recommended method and calculations, described in it are good examples and can serve as guidelines to seafarers regarding how important it is to distinguish the lights on board. It is widespread acceptance that lights in ships' compartments need to light permanently for safety reasons. Yet implementation of the above algorithm meets all requirements of MARPOL, Annex VI, and is in compliance with safety precautions on board. As the IACS members do not have requirements regarding light automation, similar calculations made on board every existing vessel would improve calculations in her Shipboard Energy Efficiency Management Plan. In this respect, all entries in SEEMP, part III, will meet MEPC requirements and guidelines regarding regulations entered into force in November 2022.

The article shows that it is more feasible for the lights in the superstructure and on the car decks to work only when people are present, even though it is not the practice in many car carriers and other ships at all. Mounting motion sensors and relays to control the lights, along with the necessary software and algorithms considerably reduce the working hours of the lights and therefore increases the energy efficiency on board all types of vessels.

The calculations presented in this article show that implementation of the proposed automation method saves about 20% of the daily energy used during navigation.

**Author Contributions:** Conceptualization, D.D.; methodology, D.D. and B.B.; data collection, D.D.; validation, B.B.; formal analysis, D.D. and B.B.; data curation, D.D.; writing—original draft preparation, D.D. and B.B.; internal review, B.B. All authors have read and agreed to the published version of the manuscript.

**Funding:** This research received no external funding.

**Informed Consent Statement:** Not applicable.

**Data Availability Statement:** Restrictions apply to the availability of these data. Data was obtained from a private pure car carrier and are available from the authors with the permission of the pure car carrier operator.

**Conflicts of Interest:** The authors declare no conflict of interest.

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
