# Peer review of "Reducing Ship’s Energy Consumption through Accommodation and Cargo Spaces Lights Automation"

_jmse, doi:10.3390/jmse11020434_

Round 1

Reviewer 1 Report

This article examines an example of using LED in a car carrier and calculates the costs saved. Although the calculation procedures, as well as the results, are plausible, there are no explicit conclusions about how the calculation approach can be usefully utilized in other vessels and other ship types. In this sense, the reviewer considers the paper in its current shape unqualified for publication in a scientific journal.

Author Response

Dear Reviewer, thank you very much for your remark.

Our answer is as follow:

This approach in ship automation can be utilized in all types of PCC vessels, where the lights are one of the large energy consumers during navigation. It can be particularly useful to owners, who want to retrofit their ships with LED lights and replace the existing fluorescent lights framework, or simply to further reduce to cost of lighting. This approach however will not be as cost effective if applied to other types of vessels, such as bulk carriers, tankers, or container ships.

Reviewer 2 Report

The authors put forward a scheme to reduce the energy consumption of ships by means of the lights automation. It is an interesting study. Although the feasibility and rationality of the proposed scheme are proved by a large number of calculation processes, there are still some problems to be solved.

1.     Personally, I think the paper lacks academic quality, and whether it meets the requirements of JMSE.

2.     Literature analysis is very weak and must be improved. The introduction and research of lights automation are not enough. It is suggested that the characteristics or benefits of LED technology on shore or in port can be increased

3.     Each calculation equation should a serial number.

4.     Tables 3 and Tables 4 should be analyzed and interpreted. For example, in line 307, how to get 80.64 kWh and 20.08 kWh should be given.

5.     It can be shown in Table 3 and Table 4 that energy consumption is greatly reduced after the use of lights automation. However, it is necessary to discuss in detail the shortcomings of lighting automation, such as the increased cost of ship and the impact of automatic system failure.

6.     How to get 104.64 kWh in line 330?

7.     Whether the lights automation transformation scheme proposed by the author conforms to the classification society's specifications should be considered, which involves the rationality of the research. The author should look up relevant documents and explain them in the paper.

Author Response

Dear Reviewer, thank you very much for your comments and feedback.

Our answers to your comments point by point are as follow:

  1. Ship's energy efficiency is part of our navigation department syllabus and summarizes our experience at sea. The results and conclusions presented in the manuscript, are discussed with our students during the course of our discipline Marine Environmental Awareness. Answering to your remarks, we will try to do our best to improve the academic and scientific quality of the manuscript.
  2. Our literature review and research focus is primarily on the existing LED lighting on board ships and its' automation in particular. We have a list of many research scientific articles discussing the topic of LED lighting in port and shore-based facilities. Reference numbers 2 and from 18 to 22 are additionally included in the manuscript.
  3. It is done.
  4. The total amount of used electricity for light fixtures in the more commonly visited rooms and stores in the superstructure for a 24-hour period without the proposed automation is the sum of the energy usage in the compartments in table 4: 11.52+3.84+0.48+1.92+7.68.....=80.64 kWh. The total amount of used electricity for the same light fixtures used for a 24-hour period with the proposed automation is the sum of the energy usage in the compartments in table 3: 3.84+0.64+0.02+...= 20.08 kWh, which is a 75% reduction in energy use. The disambiguation of this reduction can be viewed in Figure 8.
  5. The proposed equipment is widely available on the market and we don't think that ship’s cost will increase dramatically. Taking this idea, we're planning to extend our research in this field.
  6. Look at equation (1) on line 213.
  7. No legal documents state that lights should be switched on for 24hrs. All documents state that when there are workers in the compartment the lighting should be adequate and sufficient. We searched hard in the rules of the major classification societies requirements regarding LED lights automation but we couldn’t find any. As per IACS members rules are directed to the nature of the lights and their intensity but not to the automation process of turning on and off. Amendment regarding new regulations in respect of MEPC requirements, accepted by IACS members, is done (look at lines 77-84).

Reviewer 3 Report

The authors in the article proposed the use of LED lighting with automation (switch) in the superstructure and on the decks of a car carrier during regular cruises. The topic of using LED lighting on ships is well-known. For example, some large passenger ships use LED lighting throughout the ship. Therefore, the technical innovation, let alone the scientific value in the article is not high. Maybe the comments below will help the authors:

1.       The authors in the article state that the use of LED lighting will increase the energy efficiency of the ship. The EEDI index (Energy Efficiency Design Index) is used to determine the energy efficiency of newly built ships. Achieving the right EEDI is a major challenge for designers. It would be appropriate to determine the EEDI for the system without and with the solution proposed in the article, which involves changing or not changing the power of the generator sets. Such an analysis will allow additional conclusions.

2.       The input capacitor in the power circuit of the LED luminaire will generate a very large inrush current due to the charging effect when the power is switched on. This will adversely affect the reliability of the lighting system. The authors propose automatic on/off switching. How will this phenomenon be reduced.

3.       It is not clear from the article with what kind of motion sensors are used (IR, PIR, ultrasonic active sensors, camera-based, laser-based or mixed)? This is relevant to the safe operation of the vessel and crew.

4.       Most ship classification societies are not opposed to the use of LED lighting (without automation), but how will the system proposed by the authors with automatic on/off be approved by classification societies in terms of operational and crew safety?

5.       The ship's electrical network is 99.9% isolated network - there is no N. Figure 1 should be improved while adding overload/short-circuit protection to make it readable

6.       The algorithms for switching off the light would be better presented in block form.

7.       In the abstract and conclusion, the authors' achievements should be clearly highlighted because, as mentioned, LED lighting in marine applications is already in use.

8.       The authors write that the literature on the subject is small.  However, it should be increased (the possibilities are large).

Author Response

Dear Reviewer, thank you very much for your comments and feedback.

Our answers to your comments point by point are as follow:

  1. Our proposal is targeted toward existing ships, for when they decide to retrofit their lighting framework. Comments regarding MEPC and MARPOL requirements are done in the light of SEEMP and last MEPC requirements in this respect.
  2. The focus of the engineering effect of our automations it is not the topic of our research but it gives ideas of other areas to improve our research.
  3. Motion sensor type checked and amendment is done (look at line 264).
  4. No legal documents state that lights should be switched on for 24hrs. All documents state that when there are workers in the compartment the lighting should be adequate and sufficient. We searched hard in the rules of the major classification societies requirements regarding LED lights automation but we couldn’t find any. As per IACS members rules are directed to the nature of the lights and their intensity but not to the automation process of turning on and off.
  5. It is done. The image is changed. 
  6. It is done.
  7. Amendments are done in respect of your remarks in the abstract and in conclusions.
  8. It is done. Another six references are added to the manuscript.

Reviewer 4 Report

The presented work is interesting, topic is quite relevant. I have, however, some suggestions and remarks:

1. In lines 34-38 it said that the pollution is expected to increase by 50% in best case scenario and by 250% in worst case and the next sentence talks about conlcusion from marpol and co2 emissions. The paragraph is not clear to the reader, if the calim about polltion from marpol? Please provide refences and refrase the sentences. 

2. In page 3 there is long list of locations in the ship and descriptions of use of lights in those locations. Yet considering the large amount of locations it is difficult for the reader visualise the distribution of comparments. Consider adding a  blueprint or a schematic picture of a car carrier with a numbered locations in correspondance with the text. This would make it much clearer for the readers.

3. The description of lights in every location of the ship could also be restrcutured into a table with the main parameters, like compartment, number of light, power of light, operation duration and remarks. 

4.  In table 3 - 4 the reference to own research is unnecesary. The same goes for figure 2-4

5. The ships analysed in the publications are car carriers. Is there a spefici reason for selecting car carriers, if they use more light than RoRo, or other vessels it should be explained in the publication.

Author Response

Dear Reviewer, thank you very much for your comments and feedback.

Our answers to your comments point by point are as follow:

  1. It is done. References in this respect are added.
  2. It is done. Schematic picture is added and researched compartments were numbered. The names of the compartments are written on the image.
  3. It is done. A new table No 1 is done.
  4. It is done. All marks "Source" are deleted.
  5. Since the research collected information from the first author was during his regular voyage on board of PCC, the presented data refer to a car carrier. the comparative analysis is the subject of a subsequent publication.

Round 2

Reviewer 1 Report

The reviewer noticed the changes in the revised manuscript. However, no significant improvements are observed.

Author Response

Dear Reviewer,

Thank you for your comments.

The manuscript was revised and amended as per your comments. English language was revised too.

Reviewer 2 Report

The authors have responded to my comments.

Author Response

Dear Reviewer,

Thank you for your feedback. 

Reviewer 3 Report

1. after all, one could calculate the EEDI for a potentially twin designed ship which would be a good indicator of energy efficiency of the article's topic

5. the Fig. further is not clear because you connect the DC power supply to the AC terminals

Author Response

Dear Reviewer,

Thank you for your additional comments and feedback.

Our answer is:

  1. Thank you for your idea. Our next research is focused on the application of MEPC guidelines for SEEMP, Part III, and present research is good starting point.
  2. The relay on fig. 2 is replaced with universal one.